# Post-Ischemic Brain Neurodegeneration in the Form of Alzheimer’s Disease Proteinopathy: Possible Therapeutic Role of Curcumin

**DOI:** 10.3390/nu14020248

**Published:** 2022-01-07

**Authors:** Ryszard Pluta, Wanda Furmaga-Jabłońska, Sławomir Januszewski, Stanisław J. Czuczwar

**Affiliations:** 1Laboratory of Ischemic and Neurodegenerative Brain Research, Mossakowski Medical Research Institute, Polish Academy of Sciences, 02-106 Warsaw, Poland; sjanuszewski@imdik.pan.pl; 2Department of Neonate and Infant Pathology, Medical University of Lublin, 20-093 Lublin, Poland; wm.jablonska@gmail.com; 3Department of Pathophysiology, Medical University of Lublin, 20-090 Lublin, Poland; stanislaw.czuczwar@umlub.pl

**Keywords:** brain ischemia, neurodegeneration, curcumin, neuroprotection, amyloid, tau protein, dementia

## Abstract

For thousands of years, mankind has been using plant extracts or plants themselves as medicinal herbs. Currently, there is a great deal of public interest in naturally occurring medicinal substances that are virtually non-toxic, readily available, and have an impact on well-being and health. It has been noted that dietary curcumin is one of the regulators that may positively influence changes in the brain after ischemia. Curcumin is a natural polyphenolic compound with pleiotropic biological properties. The observed death of pyramidal neurons in the CA1 region of the hippocampus and its atrophy are considered to be typical changes for post-ischemic brain neurodegeneration and for Alzheimer’s disease. Additionally, it has been shown that one of the potential mechanisms of severe neuronal death is the accumulation of neurotoxic amyloid and dysfunctional tau protein after cerebral ischemia. Post-ischemic studies of human and animal brains have shown the presence of amyloid plaques and neurofibrillary tangles. The significant therapeutic feature of curcumin is that it can affect the aging-related cellular proteins, i.e., amyloid and tau protein, preventing their aggregation and insolubility after ischemia. Curcumin also decreases the neurotoxicity of amyloid and tau protein by affecting their structure. Studies in animal models of cerebral ischemia have shown that curcumin reduces infarct volume, brain edema, blood-brain barrier permeability, apoptosis, neuroinflammation, glutamate neurotoxicity, inhibits autophagy and oxidative stress, and improves neurological and behavioral deficits. The available data suggest that curcumin may be a new therapeutic substance in both regenerative medicine and the treatment of neurodegenerative disorders such as post-ischemic neurodegeneration.

## 1. Introduction

Today ischemic stroke is a huge and growing health challenge in the world. In developed and developing countries, post-ischemic brain neurodegeneration is becoming more common in view of the progressive aging of the world’s population. Ischemia-reperfusion brain neurodegeneration in the human population is the third cause of disability, second most common cause of dementia, and second cause of death worldwide, and may soon be the leading cause of dementia [1,2,3,4,5]. Eighty-four percent of ischemic stroke patients in developing countries die within three years after stroke, compared with 16% in developed countries [6]. In the population of 1 million annually, 2.4 thousand people will have an ischemic stroke and less than 50% of them will be independent a year later [5,6,7]. The epidemiological data show that annually about 17 million patients suffer from various types of cerebral ischemia in the world, and 6 million of them die each year as a result of brain ischemia [4,5,8]. Currently, it is estimated that the number of people after brain ischemia around the world reaches about 33 million [4,5,8]. According to new prognosis, the number of ischemic stroke patients in the world will increase to 77 million in 2030 [5,8]. In 2010, the annual cost of treating and caring for ischemic stroke patients in Europe was estimated at around € 64 billion [4]. In the UK, treating stroke and loss of productivity result in a social cost of £ 8.9 billion per year, with care costs accounting for about 5% of the total cost of the National Healthcare System [9]. Post-stroke neurological deficits are usually not the main problem, but the gradual, progressive cognitive decline that is associated with increased care of these patients is becoming a problem. In ischemic stroke survivors, the incidence of dementia after the first ischemic episode is estimated to be around 10% and around 41% after the next ischemic stroke [4,5]. In long-term studies of dementia after ischemic stroke, the estimated development of dementia was approximately 48% during 25 years of survival [4,8]. When the ischemic stroke trend continues, approximately 12 million patients will die by 2030, 70 million patients will be stroke survivors, and more than 200 million years of life with a disability will be recorded worldwide each year [4,8].

It is now well known that post-ischemic brain neurodegeneration is caused by a set of genetic and proteomic changes that lead to neuronal death in an amyloid- and tau protein-dependent manner, with progressive inflammation resulting in brain atrophy with the development of full-blown Alzheimer’s disease dementia [4,10,11]. Research indicates that after ischemia the brain can develop the typical neurodegeneration of Alzheimer’s disease [3,4,11,12,13,14,15,16,17,18,19,20,21,22]. First, post-ischemic brain damage causes selective neuronal death in the hippocampus typical of Alzheimer’s disease with progressive brain atrophy [23,24,25,26,27,28]. Second, neuroinflammatory changes play a key role in the progression of post-ischemic brain neurodegeneration [10,12]. Third, evidence shows that cerebral ischemia in animals and humans induces the production and accumulation of amyloid in the form of amyloid plaques [26,29,30,31,32,33,34]. Fourth, research indicates that hyperphosphorylation of the tau protein with the final formation of neurofibrillary tangles also plays a key role in the development of post-ischemic brain neurodegeneration as in Alzheimer’s disease [35,36,37,38,39,40,41,42,43,44,45,46,47,48,49,50,51,52,53,54]. Fifth, dysfunction of the autophagy, mitophagy, and apoptosis genes is involved in post-ischemic neurodegeneration, as in Alzheimer’s disease [55,56,57]. Sixth, cerebral ischemia is believed to be a causative factor in Alzheimer’s disease development [23]. It is also believed that the signaling pathways generated by amyloid and tau protein after cerebral ischemia play a decisive role in the development of irreversible neurodegeneration [26,29,35,53,54,58,59,60].

There are no therapies to prevent progressive changes from cerebral ischemia and/ or to delay or stop post-ischemic neurodegeneration. In the absence of translational experimental post-ischemic therapies in animals for clinical use [61], the emphasis is on reducing the neurotoxic effects of amyloid and tau protein on post-ischemic neurons to prevent brain neurodegeneration with dementia of Alzheimer’s disease-type. This work also focuses on the neuroprotective effects of curcumin’s pleiotropic properties on the ischemic brain.

## 2. Search Criteria and Data Collection

Published scientific papers on the use of curcumin have been screened for in vivo, in vitro, experimental and clinical studies, interactions between curcumin and the gut microbiota and vice versa, and side effects. Searches were performed digitally using databases, including PubMed, SCOPUS, MEDLINE, Science Direct, and Google Scholar to identify peer-reviewed original articles and reviews over the past two decades (1 January 2001–1 July 2021). The search strategy was carried out using the following key words: “curcumin and brain ischemia”, “brain ischemia and curcumin”, “curcumin and stroke”, stroke and curcumin”, “curcumin, neuroprotection, and brain ischemia”, “curcumin and amyloid”, “curcumin and tau protein”, “curcumin and gut microbiota”, “gut microbiota and curcumin”. A total of 1201 original papers and reviews were found, and 150 publications closely related to the subject of the review were used.

## 3. Curcumin

The favorable effects of many miscellaneous substances have been discovered through the regular consumption of plants and fruits as food. This is also the case with curcumin, whose action has been known for years in eastern countries such as India and China. Curcumin is widely used in Chinese and Indian cuisines, but has only recently been recognized as a natural remedy with proven pharmacological properties. It was chemically isolated for the first time over 200 years ago, and its structure was characterized in 1910 [62]. Curcumin is a phytopolylphenol pigment (1E,6E)-1,7-bis(4-hydroxy-3-methoxyphenyl)hepta-1,6-diene-3,5-dione) obtained from the *Curcuma longa* plant (Figure 1). *Curcuma longa* is an herbaceous perennial plant with oblong palm-like roots and tubers that grows spontaneously in Africa and South Asia, in regions with tropical climates with high rainfall. The world leader in the production of curcumin is India. Curcumin is yellow in color and is used for health, as a food preservative, but also as a fabric dye.

*Curcuma longa*, commonly known as turmeric and one of its biologically active ingredients, curcumin, is enjoying increasing clinical interest worldwide due to growing evidence of therapeutic potential resulting from numerous observations that include antioxidant, anti-inflammatory, and neurotrophic effects [63]. Curcumin is commonly used as a healing herb to aid digestion and as a culinary seasoning. With health care professionals using curcumin for a variety of clinical uses, as well as increasing interest in turmeric among laypeople, it fuels the global curcumin market. Clinical studies are heterogeneous due to the use of various forms of curcumin, turmeric essential oil, a mixture of curcuminoids, turmeric extracts, or powdered turmeric rhizome. Thus, curcumin has shown some potential in disorders, such as dermatological, gastrointestinal and neurological diseases, diabetes, cancer, and gut microbiota control [64].

## 4. Curcumin and Neuroprotection

Following experimental local brain ischemia, curcumin reduced both infarct volume and brain edema, prevented blood-brain barrier permeability, and improved neurological outcomes (Table 1) [65,66,67,68,69,70,71,72,73,74,75,76,77,78]. Curcumin had a beneficial effect on locomotor, motor, and sensory functions, as well as cognitive deficits (Table 1) [67,71,72,76,78,79,80]. Curcumin reduced neuronal apoptosis by increasing Bcl2 protein levels and by down-regulating caspase-3 mRNA and ultimately stimulating neurogenesis (Table 1) (Figure 2) [68,76,77,79,80,81,82,83,84,85,86]. Curcumin improved cerebral blood flow in the brain after ischemia by preventing neutrophil adhesion to the cerebral circulation, resulting in improved microcirculation in the brain (Table 1) [70,72]. In addition, it reduced astrogliosis and post-ischemic neuroinflammation (Table 1) [66,69,72,79]. Supplementation with curcumin reduced the inflammation of the nervous system by reducing tumor necrosis factor α, interleukin 6, and inducible nitric oxide synthase, and decreased the activity of autophagy through PI3K/Akt/mTOR [76,78,83,85].

Curcumin is also protective in rats with spontaneous hypertension prone to stroke, delaying the onset of stroke and increasing survival by improving vascular endothelial function [87]. These effects are most likely due to the increase in the presence of proteins from the family of mitochondrial anion carriers and the curcumin-induced regulation of the production of reactive oxygen species in the mitochondria [87]. These observations were also confirmed in the HUVEC cell model, using H_2_O_2_ to induce oxidative stress in vitro, which was alleviated by curcumin treatment [87]. Data similar to those obtained in the rat were also demonstrated in a mouse model of focal cerebral ischemia, in which curcumin treatment reduced the volume of cerebral infarction and neuronal apoptosis and in vitro on N2a cells, possibly by limiting mitochondrial dysfunction [88]. In parallel, curcumin influenced neurological deficits and the integrity of the ischemic blood-brain barrier, decreased brain cortex infarction, mortality, and apoptosis of neurons after cerebral ischemia [82,89,90]. Moreover, curcumin reduced the neurotoxicity of glutamate in the post-ischemic hippocampus (Table 1) [82]. In a new study, curcumin protected ischemic neuronal cells from apoptotic death through the neuroprotective effects of curcumin associated with the reciprocal inhibition of hypoxia-inducible factor-1α and autophagy (Table 1) [83].

## 5. Curcumin and Amyloid

Amyloid is a product of the metabolism of the amyloid protein precursor. The production of amyloid is catalyzed by two enzymes, β- and γ-secretase. It has been suggested that after ischemia, the development of inflammation increases the production of amyloid via increased activity of β-secretase [91]. Curcumin, by inhibiting the action of β-secretase, thus reduced the production of amyloid [91,92,93]. In addition, curcumin inhibits the maturation of the amyloid protein precursor resulting in a reduction in amyloid in brain tissue [94,95]. Curcumin may also affect amyloid production by inhibiting glycogen synthase kinase-3β mediated presenilin 1 activity, which is one of the important components of γ-secretase [96]. It has been documented that the generation of amyloid can be limited by the metal chelation phenomenon [97] and the decline in β-secretase activation by pro-inflammatory factors [58,93]. Additionally, it was presented that curcumin significantly increased the retention of the immature amyloid protein precursor in the reticulum [58]. In addition, it has been suggested that curcumin may affect the endocytosis of the amyloid protein precursor [95].

Experimental results have shown that curcumin binds to β-amyloid peptide 1–42 fibrils [98]. Curcumin has a strong inhibitory effect on amyloid aggregates in vitro, indicating that it is one of the most promising anti-amyloid substances [99,100,101,102]. In vivo and in vitro studies have revealed that curcumin has a dose-dependent effect on the inhibition of fibril development of the amyloid peptide 1–42 and 1–40, with an EC50 of 0.09–0.63 µM [92,103]. Several studies have presented that curcumin can inhibit the deposition of amyloid peptide 1–42 and 1–40 as oligomers, as well as the development of their fibril form [92,103,104]. Curcumin prevents the toxicity of the β-amyloid peptide 1–40 and inhibits the process of its aggregation [105]. The above results indicate that curcumin does not directly inhibit the development of amyloid fibrils, but rather enriches the amount of soluble oligomers and prefibrillar aggregates that do not have neurotoxic properties. The neuroprotective effect associated with curcumin is manifested in a decrease in the permeability of the cell membrane caused by amyloid. Furthermore, curcumin exerted a neuroprotective effect on amyloid-triggered toxicity by at least two compatible processes, modifying amyloid aggregation to develop non-toxic aggregates and ameliorating amyloid-induced neurotoxicity, possibly by a non-specific mechanism [105]. It was found that gold nanoparticles loaded with curcumin inhibited the aggregation of the N-terminus of amyloid and were able to dissolve its aggregates [106]. Another study provided evidence that curcumin disorganized amyloid fibrils as a result of structural changes at the salt bridge site near the C-terminus of amyloid [107]. Other studies have documented that curcumin also inhibits the development of amyloid oligomers and fibrils, binds amyloid plaques, destroys existing amyloid plaques, and decreases amyloid level and its neurotoxicity in vivo [92,105,108]. In contrast, systemic curcumin supplementation to transgenic mice for one week removed and reduced the number of amyloid plaques [108]. Curcumin also reversed structural alterations in dendrites. It follows from the above that curcumin reversed the pathological effects of amyloid and its associated toxicity in transgenic mice [108].

Post-ischemic brain amyloid level depends on the balance between brain amyloid production, clearance, and from serum influx. Thus, disruption of the clearance pathways of amyloid from the brain promotes an increase in its level in the brain parenchyma. Several possible mechanisms for removing amyloid from brain have been identified, including transport of amyloid by lipoprotein receptor related protein-1 across the blood-brain barrier into plasma, subsequent breakdown of amyloid by specialized enzymes, and also through the immune system [109]. Curcumin works similar to an amyloid vaccine by binding to amyloid, allowing it to be cleared from the brain parenchyma by promoting receptor mediated amyloid clearance [92,93]. In addition, curcumin may reduce amyloid influx into the brain parenchyma from blood by blocking the receptor for advanced glycation end-products at the blood-brain barrier and by increasing the enzymatic metabolism of amyloid [92,93]. Curcumin has phagocytic-stimulating properties and increases the number of phagocytic cells around amyloid plaques and also around human brain amyloid plaques exposed to rodent microglia after death [110,111,112]. Curcumin has been documented to induce amyloid phagocytosis through microglia activation and enzymatic metabolism [112]. Curcumin also stimulates B lymphocytes to produce anti-amyloid antibodies. In summary, curcumin simultaneously blocks the flow of amyloid from the serum into the brain parenchyma and increases its flow from the brain tissue into the blood.

## 6. Curcumin and Tau Protein

Neural cells are rich in tau protein, which is used to stabilize microtubules. Another important pathology associated with folding proteins in the brain after ischemia in humans and animals are neurofibrillary tangles, which are inherently associated with excessive hyperphosphorylation of the tau protein [35,39,49,53,113,114,115]. Hyperphosphorylated tau protein, among other things, triggers oxidative stress, causes mitochondrial dysfunction, and ultimately contributes to the progressive development of brain neurodegeneration [116]. The hyperphosphorylation and accumulation of tau protein in the form of neurofibrillary tangles are regulated by several tau protein kinases, the most common of them being glycogen synthase kinase-3β and the mitogen-activated protein kinase [39,54,96,117]. Common tau protein kinases that pathologically phosphorylate the tau protein are extracellular signal-regulated kinase 2, cyclin-dependent kinase 5, S6 kinase, SAD kinase, microtubule affinity regulating kinase, protein kinase A, calcium/calmodulin II dependent protein kinase, and kinases Fyn and c-Abl. Therefore, it is believed that the effect of curcumin on tau protein kinases is an activity to prevent or slow down [118]. Curcumin has been shown to bind to neurofibrillary tangles in the brains of animals with experimental Alzheimer’s disease, resulting in inhibition of the action of prion proteins [118]. In vitro studies have shown that curcumin inhibits the accumulation of hyperphosphorylated tau protein and disintegrates its fibers [119]. Curcumin also inhibits glycogen synthase kinase-3β activity and reduces tau protein hyperphosphorylated oligomerization and tau protein dimmer development in tau protein transgenic animals [96,120]. In addition, oral curcumin supplementation with docosahexaenoic acid reduced tau protein hyperphosphorylation by inhibiting the activity of C-Jun N-terminal kinase and insulin receptor 1 substrate [120].

## 7. Curcumin Bioavailability and Gut Microbiome

There are three compelling reasons why the therapeutic potential of curcumin has yet to be realized. The first is low oral bioavailability, mainly due to its rapid metabolism, limited absorption, and rapid systemic elimination. Second, curcumin is poorly soluble in water, around 11 ng/mL, and is highly metabolized in the body [85]. Third, when curcumin is administered orally, most of it is excreted in feces due to poor absorption in the gastrointestinal tract, moreover, curcumin is inactivated in the intestinal mucosa by glucuronidation. Curcumin then undergoes reduction in the first pass effect to hexahydrocurcumin, followed by conjugation with sulphates and glucuronides in the liver [121,122], and is finally excreted in urine [85]. Pharmacokinetic studies in rodents and humans showed that the highest blood levels achieved after oral administration were 0.051 µg/mL with 12 g of curcumin in humans, 1.35 µg/mL with 2 g/kg in the rat, and 0.22 µg/mL with 1 g/kg in mice [84]. Orally administered curcumin has been shown to have a bioavailability of 1% lower than after intraperitoneal or intravenous administration [123,124]. Curcumin appears in the blood 15 min after intraperitoneal administration, 45 min later it can be found in the liver, spleen, intestines, kidneys, and brain because curcumin crosses the blood-brain barrier [123,124]. In a rodent model, the presence of the l-piperoylpiperidine alkaloid produced from black pepper fruit, an inhibitor of uridine-5’-diphosphoglucuronosyl transferase, increases the oral bioavailability of curcumin by up to 154% [123] and results in the presence of curcumin in the brain notably up to 96 h after administration [124]. The bioavailability of curcumin can be increased by administering its derivatives which exhibit enhanced biological activity and improved pharmacokinetics, one example being dimethoxycurcumin, which has a higher level of metabolic stability [125].

The problem of the low bioavailability of curcumin is currently under discussion. Despite the low bioavailability of curcumin, its use is not ruled out, as it has beneficial effects even in small doses [126,127]. The poor bioavailability of curcumin is associated with limited absorption and rapid metabolism, which results in rapid elimination from the body. It should be added that the high concentration of curcumin causes side effects related to its action [127]. In the human body, the metabolism of curcumin takes place mainly in the small intestine and liver [127,128]. The main products of curcumin metabolism are glucuronides [127,129,130]. However, the lysosomal enzyme responsible for the deconjugation of glucuronides is present in the organisms, i.e., β-glucuronidase, and one of its substrates is curcumin glucuronide [131,132]. β-glucuronidase activity increases in inflammation [133], and the development of inflammation is associated with post-ischemic brain neurodegeneration [10,12,134]. It should be assumed that the local concentrations of curcumin may differ from those found in the blood [135]. It cannot be excluded that the inflammation associated with cerebral ischemia is responsible for the increased concentration of unmetabolized curcumin in the tissues or organs of the affected individuals. The glucuronidation of curcumin is just one of many factors affecting its bioavailability. Blood and tissue levels are affected by other factors, such as low food levels of curcumin and its interactions with other food ingredients. Curcumin is able to pass the blood-brain barrier but the permeability for the curcumin by the barrier is limited [92,127,136,137]. Although the concentration in the brain parenchyma is lower than in the blood, curcumin alleviates inflammation in the nervous system. It is currently suggested that the actual activity of curcumin in the body is not direct, but is mediated through the gut microflora [138,139]. In addition, there is evidence that gut bacteria produce large amounts of β-glucuronidase, which can raise curcumin levels [140]. This suggests that the gut microflora may control curcumin metabolism and bioavailability in the body. The gut microbiome changes throughout life [141] and progressive aging is associated with reduced microbial diversity in composition, quantity and quality, and the occurrence of cerebral ischemia [13,142,143,144,145]. There are experimental indications that curcumin may modulate the composition of the gut microbes, including microbial diversity [86,146,147,148,149]. It is believed that by modulating the gut microbiome, curcumin can reduce some of the negative consequences of post-ischemic neurodegeneration in the brain, for example by slowing it down. In conclusion, curcumin, by influencing the gut microflora, can positively affect some pathological changes. The gut microflora, through its ability to metabolize curcumin, can regulate its bioavailability.

## 8. Conclusions

Injury and death of neuronal cells, with the accumulation of diffuse amyloid and senile plaques, the development of neurofibrillary tangles, as well as neurological deficits with the development of full-blown dementia are the main phenomena in post-ischemic brain neurodegeneration in animals and humans. Due to the pleiotropic action of curcumin, such as anti-inflammatory, anti-oxidant, anti-amyloid, anti-dysfunctional tau protein, and anti-dementia properties, curcumin is a promising candidate for the prevention and therapy of post-ischemic brain neurodegeneration (Figure 2, Figure 3 and Figure 4). In addition, it is a safe substance, approved in the Europe and US as a pro-health substance, commercially available, and inexpensive. Recapitulating, the information available in this article about the pharmacological activity of curcumin provides significant evidence for the potential clinical utility of curcumin in the therapy of neurodegenerative phenomena with accumulation of folding proteins, such as amyloid and tau protein following ischemia (Figure 2).

In this review, we presented the neuroprotective effects of curcumin in post-ischemic brain neurodegeneration. Evidence shows neuroprotective, neurological, and cognitive positive effects of curcumin after ischemia-reperfusion brain injury (Figure 2). Based on the evidence presented, it seems that curcumin has therapeutic potential through anti-amyloid, anti-tau protein hyperphosphorylation, antioxidant, anti-inflammatory, anti-apoptotic effects, and influences autophagy, clearly indicating that curcumin can be used as a neuroprotective agent in post-ischemic neurodegeneration (Figure 2). It is clear that curcumin induces neuroprotection and neurogenesis, and may be a new drug substance in the treatment of aging and neurodegenerative diseases, including neurodegeneration following cerebral ischemia [150]. For this reason, curcumin may be a promising substance to counteract ischemic neurodegeneration in the future. Overall, there is a scientific rationale for the use of curcumin in the prevention and therapy of post-ischemic neurodegeneration. Nevertheless, despite preliminary hard data, prospective studies are needed to further elucidate how curcumin may be protective against ischemic brain injury and how it can be used during treatment of post-ischemic neurodegeneration. In particular, evidence from randomized controlled clinical trials will be helpful.

## 9. Outlook

The data presented in this review show a promising protective effect of curcumin after ischemia-reperfusion brain injury. However, a limited number of investigations after brain ischemia provide evidence of low or very low quality. Since post-ischemic observations have been brief, the long-term effects associated with curcumin use are currently unknown. Future randomized clinical studies are needed to confirm curcumin effectiveness and to provide additional information on some of the unresolved issues, such as how long curcumin can be used. Despite very scarce research, the results of curcumin in treating cerebral ischemia so far appear interesting in preventing the deposition of amyloid plaques and dysfunctional tau protein (Figure 3 and Figure 4). In recent years, curcumin’s reputation for pharmacological effects has steadily increased. Due to the fact that curcumin, similar to many other natural molecules, has more than one drug target, it indicates its versatile use and low risk of resistance to therapy. There is no doubt that, due to the preclinical results, the next step must be the study of curcumin in well-designed and controlled clinical trials. Double-blind studies are needed to elucidate the curcumin treatment properties. A definitive explanation of curcumin’s healing properties may offer hope for a long-term therapeutic effect. Curcumin has not been approved for clinical use. Low bioavailability is a major limitation on the utility of curcumin in the clinic. We hope that future clinical research will help us better understand the therapeutic potential of curcumin and put this fascinating molecule at the forefront of new neuroprotective therapies.

## Figures and Tables

**Figure 1 nutrients-14-00248-f001:**
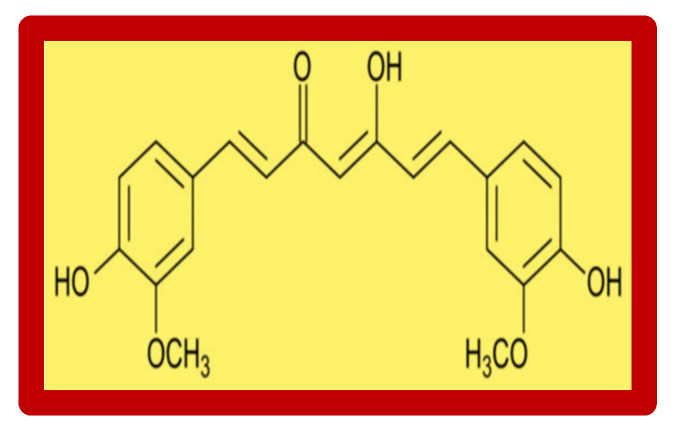
Chemical structure of curcumin.

**Figure 2 nutrients-14-00248-f002:**
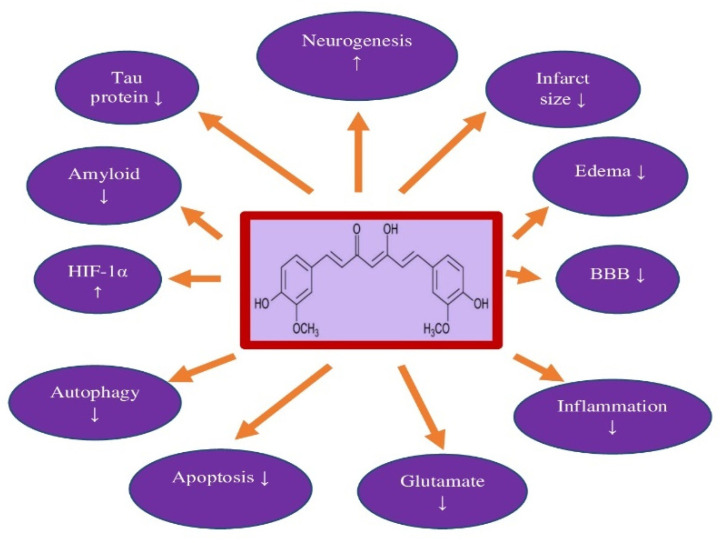
Protective influence of curcumin on post-ischemic brain neurodegeneration phenomena. In a rectangle-structure of curcumin, ↓—decrease, ↑—increase, BBB-blood brain barrier, HIF-1α-hypoxia-inducible factor-1α.

**Figure 3 nutrients-14-00248-f003:**
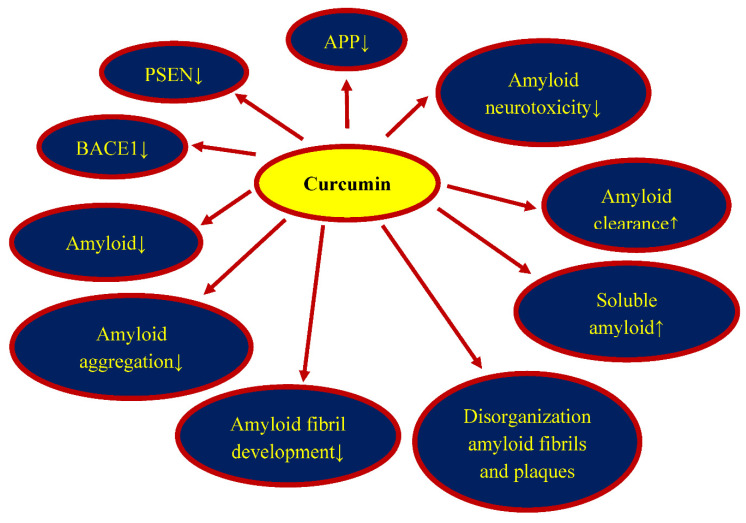
Protective effects of curcumin on post-ischemic amyloid pathology. APP-amyloid protein precursor, PSEN1-presenilin 1, BACE1-β-secretase, ↓—decrease, ↑—increase.

**Figure 4 nutrients-14-00248-f004:**
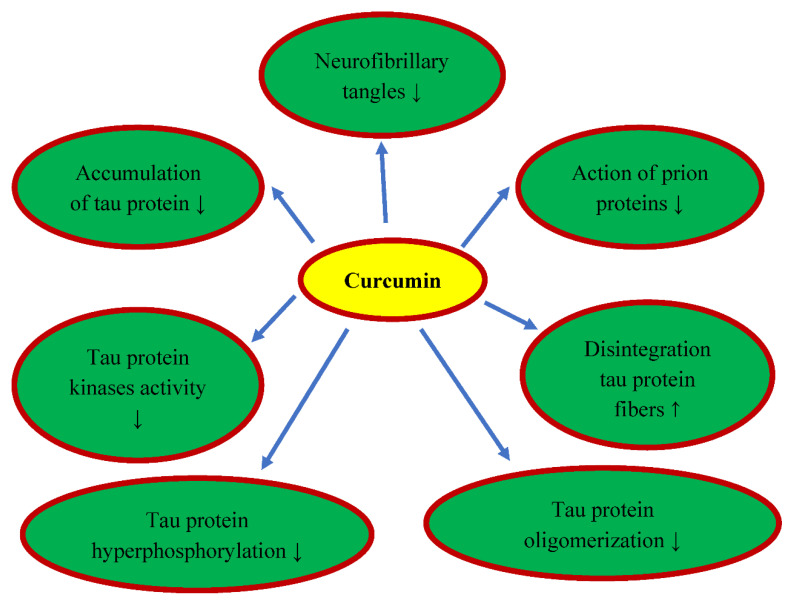
Protective effects of curcumin on post-ischemic tau protein dysfunction. ↓—decrease, ↑—increase.

**Table 1 nutrients-14-00248-t001:** Protective action of curcumin in post-ischemic neurodegeneration of the brain.

Brain Ischemia in Animals	Benefits	References
Rat, mouse	Reduction in infarct size and brain edema	[65,66,67,69,71,72,76,78]
Mouse	Reduction in the permeability of the blood-brain barrier	[66,69]
Rat, gerbil,	Decreasing apoptosis	[76,77,78,79,80,81]
Rat, mouse	Improvement of microcirculation in the brain	[70,72]
Gerbil, mouse	Reduced neuroinflammation	[72,79]
Rat	Attenuation of glutamate neurotoxicity	[82]
Rat	Mutual inhibition of hypoxia-inducible factor-1α and autophagy	[76,78,83]
Rat	Inhibition of oxidative stress	[81]
Rat	Stimulation of neurogenesis	[84]
Rat, gerbil, mouse	Improving neurological and behavioral deficits	[67,71,72,76,78,79,84]

## Data Availability

Not applicable.

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
