# Peer review of "Post-Ischemic Brain Neurodegeneration in the Form of Alzheimer’s Disease Proteinopathy: Possible Therapeutic Role of Curcumin"

_nutrients, 2022, doi:10.3390/nu14020248_

Round 1

Reviewer 1 Report

Research conducted over the last 5 years has significantly expanded our understanding of the genetic basis of post-ischemic brain neurodegeneration. It is now well known that post-ischemic brain neurodegeneration is caused by a set of genetic changes that lead to neuronal death in an amyloid and tau protein dependent manner, with progressive neuroinflammation resulting in uncontrolled brain atrophy with the development of dementia. In this review, the authors present current knowledge about the dysregulation of proteins and their genes involved in the amyloidogenic processing of the amyloid protein precursor, which is associated with the produttion of the amyloid in the post-ischemic brain. The review is well written and welcome. I do however, have three comments to make in order to improve the manuscript:

1) The aim of the study should be placed at the very end of INTRODUCTION

2) Could curcumin be useful to protect against systemic inflammation ? (see, doi: 10.3390/jcm9093053)

3) Could curcumin be useful to protect against autophagy ? (see, PMID: 28523294)

Author Response

Reviewer 1. All changes in MS are in red. Research conducted over the last 5 years has significantly expanded our understanding of the genetic basis of post-ischemic brain neurodegeneration. It is now well known that post-ischemic brain neurodegeneration is caused by a set of genetic changes that lead to neuronal death in an amyloid and tau protein dependent manner, with progressive neuroinflammation resulting in uncontrolled brain atrophy with the development of dementia. In this review, the authors present current knowledge about the dysregulation of proteins and their genes involved in the amyloidogenic processing of the amyloid protein precursor, which is associated with the production of the amyloid in the post-ischemic brain. The review is well written and welcome. I do however, have three comments to make in order to improve the manuscript: Thank you. 1) The aim of the study should be placed at the very end of INTRODUCTION. The introduction has been shortened, it is more compact and the aim of MS has been modified. 2) Could curcumin be useful to protect against systemic inflammation? (see, doi: 10.3390/jcm9093053). Yes. The suggested work was quoted in MS and placed in the literature. 3) Could curcumin be useful to protect against autophagy? (see, PMID: 28523294). Yes. This information was already in the original MS version. The suggested work was quoted in the MS and placed in the literature.

Reviewer 2 Report

1. Abstract needs modifications; a good background for the studies including tau and beta amyloid must be done. The method used for data acquisition, the credible data base searched and the key words used must be included. The main results of the search must also be reported and then a good conclusion. 2. There are a lot of redundant statistics in the introduction, there is paucity of literature on Alzhiemer's disease and alzheimerogenic proteins. The justification of the work does not tally with the topic of the manuscript. 3. The main body of the work needs better schematic diagrams to clearly demonstrate the molecular pathways the authors are trying to portray.

Author Response

Reviewer 2. All changes in MS are in red. 1. Abstract needs modifications; a good background for the studies including tau and beta amyloid must be done. The method used for data acquisition, the credible data base searched and the key words used must be included. The main results of the search must also be reported and then a good conclusion. The abstract has been modified. The method of searching for articles to work is presented in paragraph 2. Two additional figures show good conclusions. 2. There are a lot of redundant statistics in the introduction, there is paucity of literature on Alzhiemer's disease and alzheimerogenic proteins. The justification of the work does not tally with the topic of the manuscript. The introduction has been shortened, it is more compact and clear and the aim of MS has been modified. 3. The main body of the work needs better schematic diagrams to clearly demonstrate the molecular pathways the authors are trying to portray. The main part of the work has been enriched with two diagrams in order to clearly show the molecular pathways of curcumin action. - Additionally, the linguistic imperfections marked by the reviewer in the text were corrected. The curcumin figure was made by us. The abbreviations in the text are currently explained. Two figures have been added to the conclusions as suggested reviewer.

Round 2

Reviewer 2 Report

Satisfactory